# Microstructure and Low-Cycle Fatigue Behavior of Al-9Si-4Cu-0.4Mg-0.3Sc Alloy with Different Casting States

**DOI:** 10.3390/ma13030638

**Published:** 2020-01-31

**Authors:** Guanyi Wang, Xin Che, Zhipeng Zhang, Haoyu Zhang, Siqian Zhang, Zhengyuan Li, Jie Sun

**Affiliations:** 1School of Materials Science and Engineering, Shenyang University of Technology, Shenyang 110870, China; Mr_WG1@163.com (G.W.); YUNPENGZHI@163.com (Z.Z.); zhanghaoyu@sut.edu.cn (H.Z.); sqzhang@alum.imr.ac.cn (S.Z.); zhengyli@sut.edu.cn (Z.L.); 2State Key Laboratory of Rolling and Automation, Northeastern University, Shenyang 110819, China; sunjie@ral.neu.edu.cn

**Keywords:** Al-9Si-4Cu-0.4Mg-0.3Sc alloy, different casting state, low-cycle fatigue, fatigue life, deformation mechanism

## Abstract

The low-cycle fatigue behavior of Al-9Si-4Cu-0.4Mg-0.3Sc alloy with different casting states was investigated by performing low-cycle fatigue tests and by means of observations and analysis with a scanning electron microscope (SEM) and a transmission electron microscope (TEM). It was found that the metal-mold cast and die-cast Al-9Si-4Cu-0.4Mg-0.3Sc alloys exhibited the cyclic stress response of strain hardening under all imposed total strain amplitudes. The cyclic deformation resistance and fatigue life of the metal-mold cast Al-9Si-4Cu-0.4Mg-0.3Sc alloy were lower than those of the die-cast Al-9Si-4Cu-0.4Mg-0.3Sc alloy. The plastic strain and elastic strain amplitudes of the metal-mold cast and die-cast Al-9Si-4Cu-0.4Mg-0.3Sc alloys were linearly related to the number of reversals to failure, which obeyed the Coffin-Manson and Basquin formulas, respectively. The results of TEM observation revealed that at all imposed total strain amplitudes, the cyclic deformation mechanisms of the metal-mold cast and die-cast Al-9Si-4Cu-0.4Mg-0.3Sc alloys were planar slip and wavy slip at the lower and higher strain amplitudes, respectively.

## 1. Introduction

Al-Si-Cu-Mg cast alloys exhibit good casting performance, corrosion resistance, and high mechanical properties, which have been widely used in various fields of production and human life [1,2]. Compared with metal casting, die-casting not only has the advantages of high productivity and high dimensional accuracy of castings [3,4], but also can significantly improve the microstructure of alloys, thereby improving their mechanical properties [5,6]. Fatigue is one of the main failure forms of engineering components of aluminum alloys [7,8]. The structural components of Al-Si series alloys work under extremely severe and complex conditions such as high stress [9,10], and the material will be in a state of plastic strain, resulting in low-cycle fatigue failure of the material [11,12]. In order to ensure the safety of components in service, it is necessary to perform a deep investigation of the fatigue characteristics of materials. Therefore, the relevant research has attracted extensive attention from many scholars. Beroual et al. [13] studied the effect of high-pressure die-casting on the microstructures of Al-Si-Cu-Mg alloy. It was found that die-casting effectively improved the morphology of the eutectic Si phase in the alloy, and the primary eutectic Si phase changed from a lamellar to a smooth circular shape. The dense dendritic θ-Al_2_Cu phases were observed in the microstructure of the die-cast alloy. Guo et al. [14] studied low-cycle fatigue performance and compression creep fatigue performance of cast Al-Si-Cu-Mg alloy at high temperatures. It was found that under the loading condition of low-cycle fatigue at 200 °C, the cyclic stress response behavior showed cycle stability and the crack initiated at the voids and defects. When the temperature increased to 350 °C, the source of the crack was not the voids and defects, and the cyclic stress response behavior of cast Al-Si-Cu-Mg alloy exhibited a continuous softening characteristic. Cai et al. [15] summarized the low-cycle fatigue behavior of Al-10.33Si-0.4Mg-(0.28Cu) alloys at 100 °C and 250 °C at 0.25%–0.4% imposed total strain amplitudes. It was found that at 100 °C, the alloy exhibited cyclic hardening except 0.25% and softening behavior at 250 °C at all imposed strain amplitudes. At the same imposed total strain amplitude, the fatigue life of the alloy at 100 °C was higher than that at 250 °C. Zhao et al. [16] established a finite element model under low-cycle fatigue loading conditions, combined the test results, and studied the effect of specimen size and loading method on the properties of the alloy. It was found that the fracture of the specimen was a result of the combined action of tensile stress and plastic fatigue damage. AA6061, AA7075, and A6063 aluminum alloys exhibited cyclic softening behavior.

So far, the research on die-casting Al-Si-Cu-Mg alloys is mostly focused on the microstructure and static mechanical behavior, but the research about the fatigue behavior of die-cast Al-Si-Cu-Mg alloy is relatively scarcer. In addition, some research has revealed that the addition of the Sc element can effectively improve the mechanical properties of Al-Si-Cu-Mg alloys [17]. Therefore, low-cycle fatigue experiments were carried out for both the metal-mold cast and die-cast Al-9Si-4Cu-0.4Mg-0.3Sc alloys (wt %) in this investigation. The effect of the casting process on the low-cycle fatigue behavior of Al-9Si-4Cu-0.4Mg-0.3Sc alloy was discussed in order to provide a reliable theoretical basis for further improving the performance of the alloy and the development of new fatigue-resistant cast aluminum alloys.

## 2. Material and Method

The materials used in this investigation were metal-mold cast and die-cast Al-9Si-4Cu-0.4Mg-0.3Sc alloys. The melting process of the alloys was performed with an SG-5-10 type crucible resistance furnace, and the melting temperature was 740 °C. For the metal-mold casting process, the preheating temperature of the mold was 240 °C, and the casting temperature was 720 °C. The metal-mold cast bars with a diameter of 12 mm were obtained and then machined into the fatigue samples with the gauge length of 10 mm and gauge diameter of 6mm. The die-casting fatigue samples with the gauge length of 10 mm and gauge diameter of 6mm were prepared with an Evo.53D horizontal cold chamber die-casting machine. The corresponding die-casting technological parameters were the injection pressure of 60 MPa, mold temperature of 200 °C, pouring temperature of 720 °C, and holding pressure of 12 s.

The strain controlled axial pull-push fatigue tests were performed at room temperature on a PLD-50 electro-hydraulic servo fatigue testing machine (Letry, Xi’an, China). The nominal imposed total strain amplitude was between 0.25% and 0.45%, and the loading waveform was a triangular wave with a cycle frequency of 1 Hz. All fatigue experiments were carried out until the cyclic stress amplitude decreased to 80% of peak value of cyclic stress amplitude during fatigue deformation, and the corresponding number of cycles was taken as the fatigue life of the alloy. In addition, at least two samples were used for the fatigue test performed at each strain amplitude.

The microstructures of the alloy were observed with a S-3400N type SEM (Hitachi, Japan), and the microstructures of the fatigue deformation zone were characterized with a TECNAI 20 type TEM. The TEM samples were prepared using the electrolytic thinning method on an MTP-1A electro polisher (Hello, Beijing, China). The composition of electrolyte was 30 mL HNO_3_ and 70 mL CH_3_OH, the double injection voltage was 15 V, the corresponding current was about 80mA, and the temperature of electrolyte was about −30 °C. 

## 3. Results and Discussion

### 3.1. Microstructure

Figure 1 illustrates the microstructures of both the metal-mold cast and die-cast Al-9Si-4Cu-0.4Mg-0.3Sc alloys. Compared with the metal-mold cast alloy, the microstructure of the die-cast Al-9Si-4Cu-0.4Mg-0.3Sc alloy is remarkably refined, and the grains of the α-Al matrix phase show equiaxial shape. Figure 2 shows the SEM images and EDAX (Energy Dispersive X-ray Analysis) analysis results for the eutectic Si phase and Al_2_Cu phase in the metal-mold cast and die-cast Al-9Si-4Cu-0.4Mg-0.3Sc alloys. It can be seen from Figure 2a that in the microstructure of the metal-mold cast Al-9Si-4Cu-0.4Mg-0.3Sc alloy, the lamellar eutectic Si phase and the irregularly shaped Al_2_Cu phase mainly distribute at the α-Al grain boundary. However, in the microstructure of the die-cast Al-9Si-4Cu-0.4Mg-0.3Sc alloy, the shape of the eutectic Si phase changed into a bulk-like shape, and its size was much smaller, as shown in Figure 2b.

### 3.2. Cyclic Stress Response Behavior

The cyclic stress response curves of the Al-9Si-4Cu-0.4Mg-0.3Sc alloy with different casting states at imposed total strain amplitudes from 0.25% to 0.45% are shown in Figure 3. It was found that at each imposed total strain amplitude, the cyclic stress amplitude of the metal-mold cast Al-9Si-4Cu-0.4Mg-0.3Sc alloy was lower than that of the die-cast Al-9Si-4Cu-0.4Mg-0.3Sc alloy. When the imposed total strain amplitude was 0.25%, the metal-mold cast Al-9Si-4Cu-0.4Mg-0.3Sc alloy showed a cyclic hardening behavior at the early stage of cyclic deformation. As the cyclic number increased, the rate of cyclic hardening gradually decreased. However, the die-cast Al-9Si-4Cu-0.4Mg-0.3Sc alloy exhibited cyclic hardening during the entire fatigue deformation, and the rate of cyclic hardening gradually decreased by increasing the number of cycles. When the imposed total strain amplitudes were in the range from 0.3% to 0.45%, the alloy with two cast states exhibited a cyclic hardening tendency during the entire fatigue deformation.

### 3.3. Cyclic Stress–Strain Behavior

Figure 4 shows the cyclic stress amplitude versus plastic strain amplitude curves for the Al-9Si-4Cu-0.4Mg-0.3Sc alloy with different casting states. It was noted that the relationship between the cyclic stress amplitude and plastic strain amplitude was a single-scope linear relationship for the alloy with different casting states. Thus, the cyclic stress versus strain behavior of the Al-9Si-4Cu-0.4Mg-0.3Sc alloy can be expressed by the following formula:(1)$$\Delta \sigma /2=K^{\prime }{\left(\Delta \varepsilon_{p}/2\right)}^{n^{\prime }}$$
where Δ*σ*/2 is the cyclic stress amplitude at half-life, *K*′ refers to the cyclic strength coefficient, and *n*′ stands for the cyclic strength hardening exponent. Through linear regression analysis, the values of *K*′ and *n*′ for the alloy with different casting states can be determined and are listed in Table 1.

### 3.4. Low-Cycle Fatigue Life Behavior

Figure 5 shows the relationship between the imposed total strain amplitude and low-cycle fatigue life for the Al-9Si-4Cu-0.4Mg-0.3Sc alloy with different casting states. It was seen that at the same imposed total strain amplitude, the low-cycle fatigue life of the metal-mold cast Al-9Si-4Cu-0.4Mg-0.3Sc alloy was lower than that of the die-cast Al-9Si-4Cu-0.4Mg-0.3Sc alloy. When the die-casting technology was applied, the low-cycle fatigue performance of Al-9Si-4Cu-0.4Mg-0.3Sc alloy could be improved.

For the low-cycle fatigue tests where the imposed total strain amplitude is controlled, the relationship among the plastic strain amplitude (Δ*ε_p_*/2), the elastic strain amplitude (Δ*ε_e_*/2), and the number of reversals to failure (*2N_f_*) can be described by Formula (2), the Coffin-Manson formula, and Formula (3), which introduces elastic modulus into the Basquin formula.
(2)$$\Delta \varepsilon_{p}/2=\varepsilon_{f}^{\prime }{\left(2N_{f}\right)}^{c}$$
(3)$$\Delta \varepsilon_{e}/2=\frac{\sigma_{f}^{\prime }}{E}{\left(2N_{f}\right)}^{b}$$
where *ε_f_*′ is the fatigue ductility coefficient, *c* refers to the fatigue ductility exponent, *σ_f_*′ stands for the fatigue strength coefficient, *b* stands for the fatigue strength exponent, and *E* is Young’s modulus.

Figure 6 shows the relationship curves between either plastic strain amplitude or elastic strain amplitude and reversals to failure for the two cast alloys. As shown in Figure 6, the curves of both plastic and elastic strain amplitudes as a function of reversals to failure for the two cast alloys were linear. With the linear regression analysis method, such strain fatigue parameters as *ε_f_*′, *c*, *σ_f_*′, and *b* for the two cast alloys were calculated, and the corresponding results are also given in Table 1. Both *K*′ and *n*′ values of the die-cast Al-9Si-4Cu-0.4Mg-0.3Sc alloy were higher than those of the metal-mold cast Al-9Si-4Cu-0.4Mg-0.3Sc alloy.

Figure 7 shows the TEM images of dislocation configuration for the Al-9Si-4Cu-0.4Mg-0.3Sc alloy after low-cycle fatigue deformation. It was seen that when the imposed total strain amplitude was higher, the dislocation configuration of the fatigue deformation zone for the two cast Al-9Si-4Cu-0.4Mg-0.3Sc alloys was mainly a cellular substructure, and a higher density of dislocations inside some dislocation cells existed. The plastic deformation of the alloy was in wavy slip mode. During the movement of dislocations, the dislocation cell walls were formed by cross-slip of dislocations. When the dislocation cell walls joined end to end, these cellular substructures were formed. However, under the lower imposed total strain amplitude, the dislocation configuration of the fatigue deformation zone was the slip bands with crossed distribution along two different directions, as indicated by the arrows in Figure 7a,c. The projections of these intersecting slip bands on plane (001)_Al_ were almost vertical [18], showing a typical planar slip mechanism.

## 4. Discussion

Compared with metal-mold casting, the microstructure of the Al-9Si-4Cu-0.4Mg-0.3Sc alloy fabricated by die-casting was remarkably refined. Generally, the superior mechanism of die-casting to refine grains is closely related to the filling of the alloy liquid at a higher speed during the die-casting process and solidification under high pressure. On the one hand, in the first stage of solidification, the alloy liquid passes through the ingate at a high speed and enters the cavity in an approximately continuous droplet flow. The alloy liquid injected into the cavity at the early stage starts to solidify after touching the cavity wall. The injected alloy liquid then impacts the solidified layer crystals at a high speed, which cause the crystals of the solidified layer to fall off. These exfoliated crystals enter the unset liquid alloy and act as a new nucleation substrate, which leads to the refinement of the grains. On the other hand, in the second and third stages of solidification, the alloy liquid solidifies under pressure. According to Clapeyron’s equation [19],
(4)$$\;dT=\frac{T_{m}\left(V_{s}{}^{\_}V_{l}\right)}{\Delta H_{m}}dp$$
where d*p* is the pressure variation, d*T* stands for the change of melting point corresponding to the change of d*p* value, *V_s_* and *V_l_* are the volume of solid and liquid phases of the alloy per unit mass, respectively, Δ*H_m_* refers to the latent heat of fusion, and *T_m_* stands for the fusion temperature. 

It can be seen that in the solidification process of the alloy, the actual melting point of the alloy liquid increases with the increasing pressure. This also means that the undercooling (Δ*T*) of the alloy liquid also increases. According to the solidification nucleation theory, the degree of undercooling, the critical nucleation radius (*r_k_*) [20], and critical nucleation energy (Δ*G^0^*) [21] have the following relationship: (5)$$r_{k}=\frac{2\sigma T_{m}}{L_{m}\Delta T}$$
(6)$$\Delta G^{0}=32\sigma^{3}{\left[\frac{dT}{\rho \Delta T\left(V_{s}-V_{l}\right)dp}\right]}^{2}$$
where *L_m_* refers to the latent heat of fusion, *ρ* is the density of the alloy, and *σ* stands for the surface tension of the alloy liquid.

It can be seen that when the alloy solidifies under the imposed pressure, the critical nucleation radius and critical nucleation energy simultaneously reduce, thereby causing an increase in the number of crystal cores. Higher amounts of crystalline cores lead to a finer structure of the alloy. In addition, under the influence of pressure, the alloy melt comes in contact more closely with the mold wall, and the heat exchange condition between the mold and alloy melt is improved. This also results in an increase of undercooling degree and promotes the nucleation and the formation of the microstructure with fine grains. In the die-casting process, the alloy melt solidifies within a very short time, and the preferential growth of the eutectic Si phase becomes very difficult. Therefore, the eutectic Si phase in the die-cast alloy changes from a lamellar to a bulk shape.

As previously mentioned, the cyclic stress amplitude of the die-cast Al-9Si-4Cu-0.4Mg-0.3Sc alloy was higher than that of the metal-mold cast Al-9Si-4Cu-0.4Mg-0.3Sc alloy. The situation was closely related to the grain refinement effect during alloy solidification in a high-pressure environment. During the fatigue deformation process, the grain boundary hinders the dislocation motion. Dislocations pile up near the grain boundaries and form a dislocation pile-up group. This dislocation pile-up group generates near the grain boundary and produces a reaction force to the source of dislocations within the grain, which is proportional to the number of dislocations piling up near the grain boundary. When the number of dislocations near the grain boundary increases to a certain value, the dislocation source inside the grains stop acting. Thus, additional stresses need to be applied to enhance the action of the dislocation source. Because the grain boundary is an obstacle to the dislocation motion, it is almost impossible for the dislocations to slip directly from one grain to the adjacent grains. In order to ensure the occurrence of dislocation slip in the adjacent grains, it is necessary to increase the applied stress to activate the dislocation source in the adjacent grains. The reduction in the grain size increases the resistance of the dislocation motion, reduces the mobility of dislocation, and finally increases the cyclic stress amplitude of the alloy.

Compared with the fatigue life of the metal-mold cast Al-9Si-4Cu-0.4Mg-0.3Sc alloy, the fatigue life of the die-cast Al-9Si-4Cu-0.4Mg-0.3Sc alloy was remarkably improved. On the one hand, the eutectic Si phase in the die-cast Al-9Si-4Cu-0.4Mg-0.3Sc alloy structure was a relatively small block, which reduced the stress concentration of the alloy, thus reducing the fatigue crack initiation and increasing the fatigue life of the alloy. On the other hand, the smaller grain size led to a higher density of the slip band formed during fatigue deformation. It also helped the plastic deformation to be activated on more slip bands, which relieved the unevenness of plastic deformation, reduced the stress concentration, and decreased the initiation of fatigue crack on the slip band. Therefore, due to the characteristics of slip deformation, reducing the grain size delayed the fatigue crack initiation and increased the fatigue life of the alloy.

## 5. Conclusions

(1) In comparison with metal-mold casting, die-casting significantly refined the microstructure of the Al-9Si-4Cu-0.4Mg-0.3Sc alloy. The eutectic Si phase showed a block-like shape in the die-cast alloy, while there was a lamellar shape in the metal-mold cast alloy.

(2) In comparison with metal-mold casting, die-casting significantly improved the low-cycle fatigue life of the Al-9Si-4Cu-0.4Mg-0.3Sc alloy. The relationship between the plastic strain and elastic strain amplitudes with the number of reversals to failure can be described by the Coffin-Manson and Basquin formulas, respectively.

(3) The cyclic plastic deformation mechanism of the Al-9Si-4Cu-0.4Mg-0.3Sc alloy with different casting states was planar slip under the lower total imposed strain amplitude. Under the higher total imposed strain amplitude, the cyclic plastic deformation mechanism of the alloy changed into the wavy slip.

## Figures and Tables

**Figure 1 materials-13-00638-f001:**
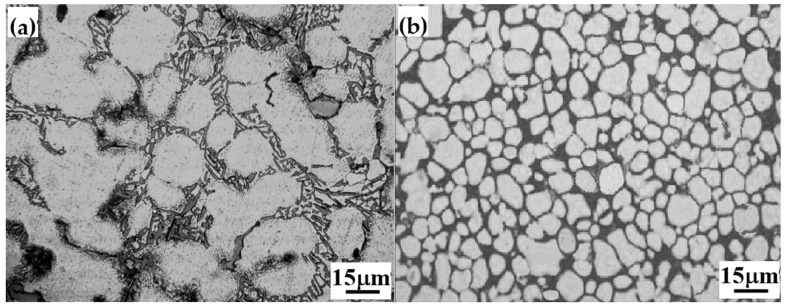
Microstructure of metal-mold cast and die-cast Al-9Si-4Cu-0.4Mg-0.3Sc alloys. (**a**) Microstructure of a metal-mold cast alloy; and (**b**) microstructure of a die-cast alloy.

**Figure 2 materials-13-00638-f002:**
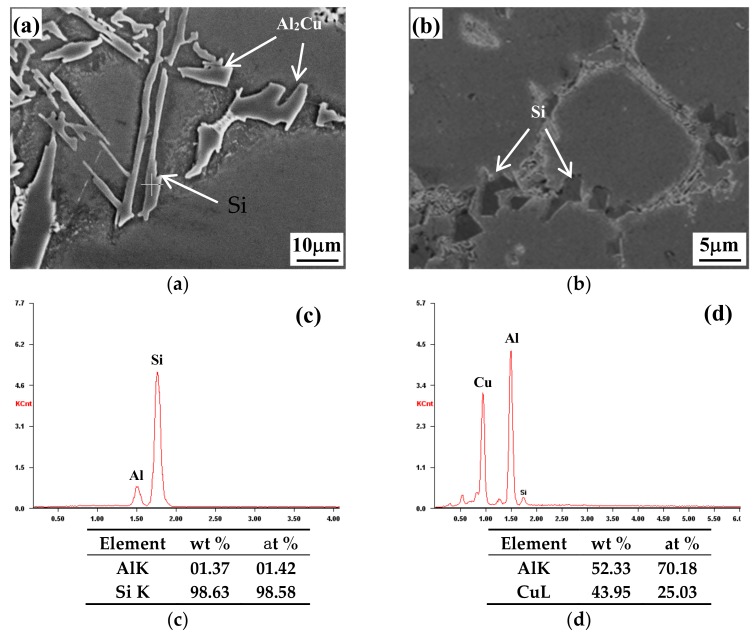
SEM image and EDAX analysis results for metal-mold cast and die-cast Al-9Si-4Cu-0.4Mg-0.3Sc alloys. (**a**) SEM image of a metal-mold cast alloy; (**b**) SEM image of a die-cast alloy; (**c**) EDAX analysis of the Si phase of a metal-mold cast alloy; and (**d**) EDAX analysis of the Al_2_Cu phase of a metal-mold cast alloy.

**Figure 3 materials-13-00638-f003:**
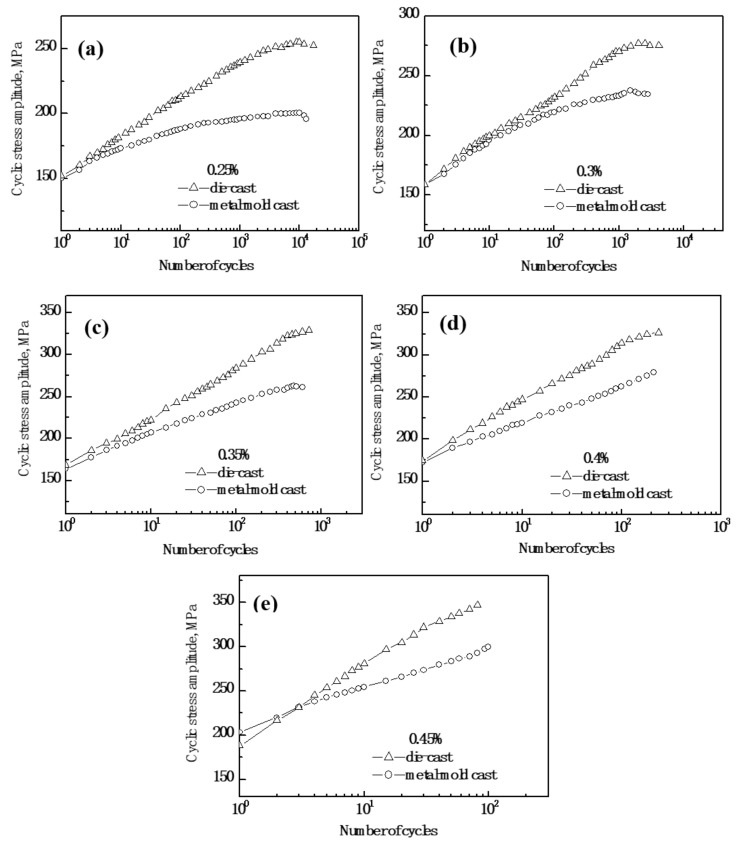
Cyclic stress response curves of metal-mold cast and die-cast Al-9Si-4Cu-0.4Mg-0.3Sc alloys at various total strain amplitudes. (**a**) 0.25%; (**b**) 0.3%; (**c**) 0.35%; (**d**) 0.4%; and (**e**) 0.45%.

**Figure 4 materials-13-00638-f004:**
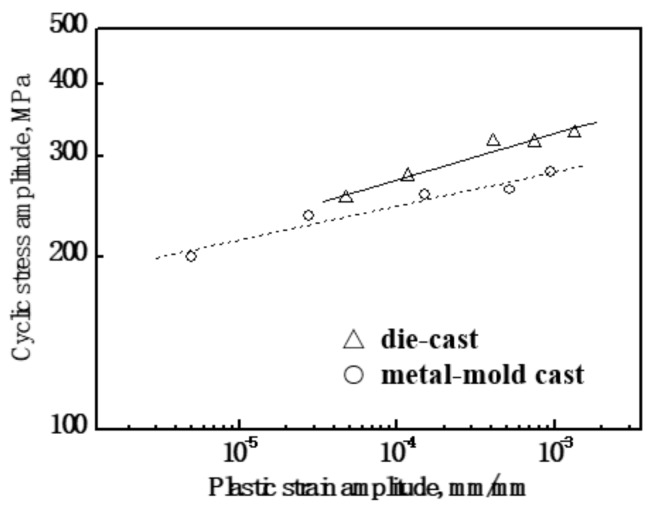
Cyclic stress–strain curves for metal-mold cast and die cast Al-9Si-4Cu-0.4Mg-0.3Sc alloys.

**Figure 5 materials-13-00638-f005:**
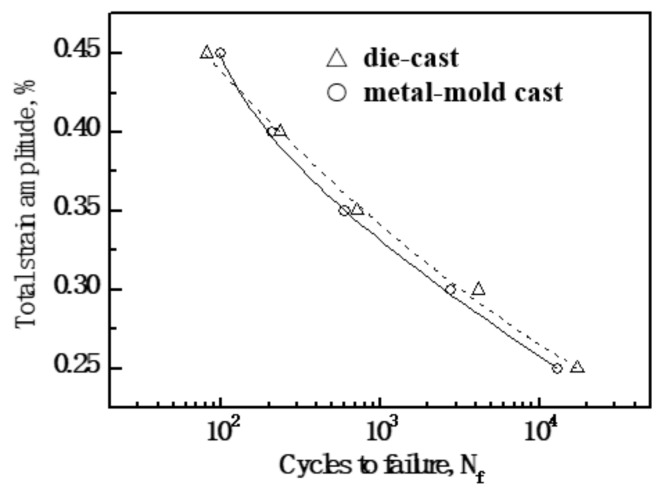
Total strain amplitude versus fatigue life for metal-mold cast and die cast Al-9Si-4Cu-0.4Mg-0.3Sc alloys.

**Figure 6 materials-13-00638-f006:**
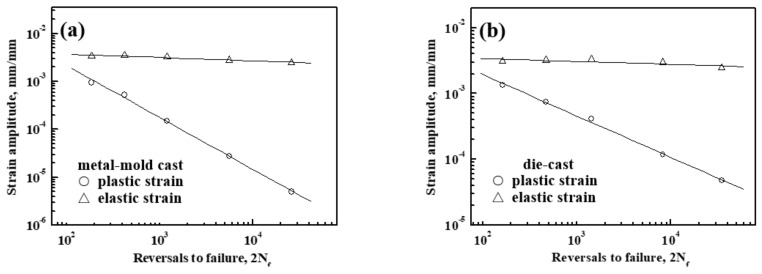
Strain amplitudes versus reversals to failure for Al-9Si-4Cu-0.4Mg-0.3Sc alloy. (**a**) metal-mold cast state; (**b**) die-cast state

**Figure 7 materials-13-00638-f007:**
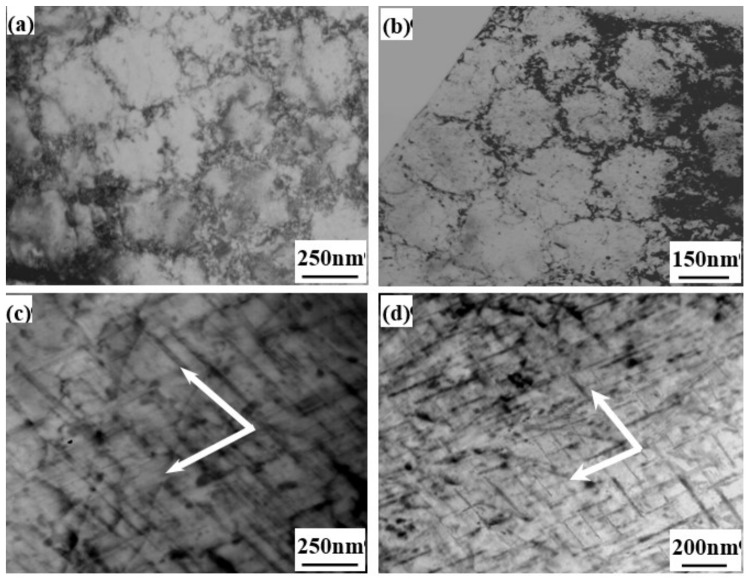
Dislocation configurations in Al-9Si-4Cu-0.4Mg-0.3Sc alloys after fatigue failure (arrows indicate the direction of slip bands). (**a**) Metal-mold cast alloy under Δ*ε_t_*/2 = 0.45%; (**b**) die-cast alloy under Δ*ε_t_*/2 = 0.45%; (**c**) metal-mold cast alloy under Δ*ε_t_*/2 = 0.25%; and (**d**) die-cast alloy under Δ*ε_t_*/2 = 0.25%.

**Table 1 materials-13-00638-t001:** Strain fatigue parameters of the metal-mold cast and die-cast Al-9Si-4Cu-0.4Mg-0.3Sc alloys.

Alloy	*K*′ (MPa)	*n*′	*σ_f_*′ (MPa)	*b*	*ε_f_*′ (%)	*c*
metal-mold cast alloy	424.2	0.059	355.2	-0.070	31.3	-1.082
die-cast alloy	569.8	0.080	284.9	-0.043	3.5	-0.629

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
