# Peer review of "Microstructure and Low-Cycle Fatigue Behavior of Al-9Si-4Cu-0.4Mg-0.3Sc Alloy with Different Casting States"

_materials, 2020, doi:10.3390/ma13030638_

Round 1

Reviewer 1 Report

The authors investigate the microstructure and low-cycle fatigue behaviour of an Al-Si-Cu-Mg-Sc alloy and compare different casting approaches. The study is interesting, sufficiently novel to deserve publication in Materials and the paper is in general well-written (only requiring minor English proof-reading). However, the authors should take care of two aspects before publication:

1) There is a vast literature on the fatigue behaviour of Al-Si-Cu-Mg alloys and some of the findings are not now (or could be expected from such literature). The authors must place their work in the context of the state-of-the-art.

2) The quality of most of the figures could be improved (particularly the ones including graphs).

Author Response

Response to Reviewer Comments

Dear reviewer(s):
         We are very grateful to your comments for the manuscript. According with your advice, we amended the relevant part in manuscript. Below are answers to the points raised. Changes are highlighted in red in the manuscript itself.

Point 1: There is a vast literature on the fatigue behaviour of Al-Si-Cu-Mg alloys and some of the findings are not now (or could be expected from such literature). The authors must place their work in the context of the state-of-the-art.

Response 1: We are very grateful to the reviewer for your suggestion on supplementary literature on fatigue behavior of Al-Si-Cu-Mg alloy. The author has read a lot of such literature, and selected a part of them as a theoretical support.

Point 2: The quality of most of the figures could be improved (particularly the ones including graphs).

Response 2: We also attach great importance to the comment of the reviewer on the need to improve the quality of the figures. The author has replaced some of the figures with high-definition figures, sharpened the figures, and removed unnecessary symbols in the figures.

Reviewer 2 Report

Review for materials- 705699

Microstructure and Low-Cycle Fatigue Behavior of Al-9Si-4Cu-0.4Mg-0.3Sc Alloy with Different Casting States

The authors present an interesting for the readers of the journal Materials. Anyway, I present several recommendations:

* Line 28, after first sentence: Please, include some example of application of this material.

* Lines 32-33: “Therefore, the relevant researches have attracted extensive attention from many scholars at home and abroad [5-7]”. Please, remove at home and abroad. In my opinion, this makes no sense.

* Figures 2 (a, b, c and d): Please, remove the following symbols ( ) when doing the screenshot.

* Please, revise the format of the references according the MDPI guidelines. Do not use superscript.

* Conclusion section: “And the relationship between between the plastic strain and elastic strain amplitudes with the number of reversals to failure can be discribed with the Coffin-Manson and Basquin formulas, respectively.” These formulas are not cited in the main text. Please, include an explanation in the main text in order to link to the Conclusion section.

* The number of references should be increased. Furthermore, it would be advisable to include some papers from the journals of MDPI editorial (Metals, Applied Sciences, Materials, etc.) related to the topic of the manuscript.

Round 2

Reviewer 2 Report

Accept in present form.